# Spurious Asthma Presentation during COVID-19

**DOI:** 10.3390/children9010005

**Published:** 2021-12-23

**Authors:** Israel Amirav, Moran Lavie

**Affiliations:** Pediatric Pulmonology Unit, Dana-Dwek Children’s Hospital, Tel-Aviv Sourasky Medical Center, Sackler Faculty of Medicine, Tel Aviv University, Tel Aviv 6423906, Israel; moranla@tlvmc.gov.il

**Keywords:** asthma, face mask, COVID-19, hyperventilation

## Abstract

The use of face masks as a means for preventing the spread of SARS-CoV-2 is now a common practice world-wide. Three children presented to our specialty clinic with respiratory complaints during protective face mask wearing. They were diagnosed as asthma and referred to our specialist clinic for further evaluation after asthma treatments were ineffective. Full details and a video clip demonstrating the effects of wearing the mask is presented for the first patient. The detailed evaluation confirmed the diagnosis of hyperventilation. *Conclusions:* In the current era of the daily use of masks, pediatricians should be aware of potential anxiety and hyperventilation while the mask is being worn, causing symptoms that mimic common respiratory disorders, such as asthma.

## 1. What Is Known

Masks are commonly used these days.

Wearing face masks may have some caveats and limitations that should be taken into account.

## 2. What Is New

Pediatricians should be aware of potential anxiety and hyperventilation while the mask is being worn, causing symptoms that mimic common respiratory disorders, such as asthma.

## 3. Case Vignette

A 16-year-old girl was referred to our asthma clinic during the COVID-19 pandemic for investigation of suspected asthma not responding to treatment.

She had a history of neurofibromatosis type 1 as well as congenital scoliosis for which she underwent spinal fusion at the age of 14. She had no significant respiratory history, and her family history was unremarkable.

The patient’s mother reported occasional episodes of dyspnea with some noisy breathing in enclosed spaces and a recent increase in frequency when the child was outdoors wearing a face mask. No coughing or wheezing was observed, and she exhibited no respiratory symptoms during sleep. Due to her dyspneic episodes, she was suspected to have asthma and was prescribed bronchodilator inhalers with no effect. She described her difficulties as mostly the inability “to breathe air in” and pointed to her throat as the source of difficulty.

In our clinic, she was observed to be breathing normally, and an examination of her lungs was unremarkable. Shortly after wearing her face mask, however, her breathing became rapid, labored and noisy with minimal retrosternal notch retractions with no change in her oxygen saturation. Repeated physical auscultation revealed minimal inspiratory stridor with no change in her voice. The symptoms resolved when the mask was taken off (see Appendix A).

The pulmonary function tests were normal both for the expiratory and inspiratory loops with no reversibility. Fractional exhaled nitric oxide was normal. The chest X-ray and lateral neck X-rays were normal. Direct laryngoscopy (through the nose without the face mask) revealed normal anatomy and function.

The working diagnosis at that point was respiratory distress or hyperventilation secondary to the use of a face mask.

## 4. Discussion

Following the outbreak of the COVID-19 pandemic, the use of face masks as a means for preventing the spread of SARS-CoV-2 was recommended by the World Health Organization (WHO) and the Center for Disease Control (CDC) [1,2]. The CDC recommends that masks be worn whenever social distancing becomes difficult, with the aim of personal protection, bearing in mind that asymptomatic individuals can also transmit the virus [3,4]. In general, face masks do not appear to significantly change breathing physiology [5]. However, face masks have some caveats and limitations that should be taken into account, particularly those related to perceived discomfort and a potential increase in end-tidal CO_2_ [6,7,8,9]. People with underlying anxiety disorders or claustrophobia can particularly be affected by wearing a mask. Many people find that it takes a few days to adjust to the mask, and the accommodation period can be longer because of the increased sense of breathlessness for those with pulmonary conditions, such as asthma and chronic obstructive pulmonary disease (COPD). Hyperventilation refers to an inappropriate increase in minute ventilation beyond metabolic needs and may be associated with several somatic symptoms, such as dyspnea, dizziness or lightheadedness, chest pain or tightness [10,11]. When hyperventilating, people breathe with their neck and chest muscles and not with their diaphragm. Hyperventilation may lower the level of CO_2_ and reduce cerebral perfusion, resulting in anxiety, confusion and potential fatigue. Respiratory sequelae associated with an acute episode of hyperventilation may include dyspnea, air hunger, the sensation of an inability to inhale a complete breath and/or a feeling of unsatisfying breathing. Patients may complain of “shortness of breath”, stating “I can’t get enough air or oxygen”, or “I can’t fill my lungs with air”, often with some vague chest discomfort and often with an anxiety component [12]. Common respiratory conditions, such as asthma, COPD and upper airway obstruction, may present with features common to hyperventilation.

Our patient exhibited no evidence of airway disease (i.e., she had normal pulmonary function tests, normal fractional exhaled nitric oxide and no response to bronchodilators). Her medical background and particularly the history of dyspnea in enclosed spaces, however, could make her vulnerable to anxiety and associated hyperventilation with the mask [13]. We referred her to training in breathing exercises, after which she reported some improvement in her unsatisfying breathing. On follow up three months later, parents reported complete resolution of her symptoms. They associated it with the general ease of public restrictions, including the waiver to strict face mask wearing.

Recently we saw two more children (a 12-year-old girl and 9-year-old old boy) who presented to our clinic with similar complaints. Careful evaluation helped exclude asthma, which assured parents of the benign nature of their child’s symptoms. It is noteworthy that problems with the mask may also occur in a patient with known asthma. In that case, if the caregivers are not careful, they may prescribe oral steroids inappropriately. In addition, a psychological assessment, as well as careful clinical assessment for intrathoracic vs. extrathoracic causes of dyspnea can aid primary care caregivers in arriving at an accurate diagnosis. Such an assessment could probably be done at the office without the need for a specialist referral.

Pediatricians should convey the message that face mask wearing is extremely important when it comes to slowing the spread of COVID-19. In addition, they should be aware of potential anxiety and hyperventilation while the mask is being worn, causing symptoms that mimic common respiratory disorders, such as asthma. Reassurance and encouragement, with easy-to-do-at-home breathing exercises (see Appendix A), can be helpful in these cases.

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
