# Peer review of "Spurious Asthma Presentation during COVID-19"

_children, 2021, doi:10.3390/children9010005_

Round 1

Reviewer 1 Report

Thank you for allowing me to review this paper. It is a presentation of cases of children with respiratory symptoms that mimics asthma. Although these children were given treatments , their response was minor so referred to the specialist clinic. The authors presented excellent case with enough details. Their final conclusion was hyperventilation. They adviced breathing exercises which showed positive effect. I would like to see some elaboration about these recommended exercises to help the reader better understand . I also would love to see more discussion on the importance of wearing the masks in helping to prevent COVID19 spreading, so readers don't get the Impression that removing the mask in such cases is better. You may elaborate about this in the final paragraph. Also provide more details about the exercises you recommended....the procedure maybe. 

Ethical approval: although authors think this was not a study, permission to include the video and the story required

References: please ensure style be followed by journal rules. Particularly the first 2 references

All the best

Author Response

Thank you for allowing me to review this paper. It is a presentation of cases of children with respiratory symptoms that mimics asthma. Although these children were given treatments , their response was minor so referred to the specialist clinic. The authors presented excellent case with enough details. Their final conclusion was hyperventilation. They adviced breathing exercises which showed positive effect. I would like to see some elaboration about these recommended exercises to help the reader better understand . I also would love to see more discussion on the importance of wearing the masks in helping to prevent COVID19 spreading, so readers don't get the Impression that removing the mask in such cases is better. You may elaborate about this in the final paragraph. Also provide more details about the exercises you recommended....the procedure maybe.

We appreciate the reviewers’ suggestions. Accordingly, we have now enclosed a supplement to describe optional breathing exercises.  We have now stressed the importance of wearing masks to prevent COVID-19 spread (final paragraph).

Ethical approval: although authors think this was not a study, permission to include the video and the story required

Thank you for this note. A consent to publish the video and story was provided to the journal’s staff upon manuscript submission. We have now included this statement in the manuscript.

References: please ensure style be followed by journal rules. Particularly the first 2 references

Thank you. The references are now reformatted to the “instructions to authors” at

Reviewer 2 Report

To the authors

SUMMARY

This study reports three pediatric case studies where the wearing of surgical face masks to protect against COVID-19 infection caused asthma episodes attributed to hyperventilation. According to the investigators, these observations were most likely caused by increased anxiety of mask-wearing which promoted asthma-like symptoms. Indeed, these observations are important and highlight that pediatricians should be informed of asthma-like symptoms in children caused by the wearing of surgical masks and not be misdiagnosed as ‘true asthma’. Furthermore, the investigators conclude by recommending that breathing exercises and continued reassurance to reduce anxiety could relieve symptoms.

This manuscript is well-written and comprehensible for readers even without a medical background. Given the current COVID pandemic, in my opinion, this topic is of great interest to health professionals, especially pediatricians working with respiratory disorders.

Q. Have these side effects of mask-wearing in children been mentioned in the literature in any other country?

I have no further comments.

Author Response

This study reports three pediatric case studies where the wearing of surgical face masks to protect against COVID-19 infection caused asthma episodes attributed to hyperventilation. According to the investigators, these observations were most likely caused by increased anxiety of mask-wearing which promoted asthma-like symptoms. Indeed, these observations are important and highlight that pediatricians should be informed of asthma-like symptoms in children caused by the wearing of surgical masks and not be misdiagnosed as ‘true asthma’. Furthermore, the investigators conclude by recommending that breathing exercises and continued reassurance to reduce anxiety could relieve symptoms.

This manuscript is well-written and comprehensible for readers even without a medical background. Given the current COVID pandemic, in my opinion, this topic is of great interest to health professionals, especially pediatricians working with respiratory disorders.

  1. Q. Have these side effects of mask-wearing in children been mentioned in the literature in any other country?

To the best of our knowledge, we have found no reported similar presentation as our case.

I have no further comments.